

# Influence of minor hydrocarbon seepage on sulfur cycling in marine subsurface sediments and its significance for hydrocarbon reservoir detection

5    Ellen Schnabel[1], Aurèle Vuillemin[1], Cédric C. Laczny[2], Benoit J. Kunath[2], André R. Soares[3], Rolando Di Primio[4], Jens Kallmeyer[1*] and the PROSPECTOMICS Consortium[+]

[1] GFZ German Research Centre for Geosciences, Section Geomicrobiology, Telegrafenberg, 14473 Potsdam, Germany

10   [2] Luxembourg Centre for Systems Biomedicine, University of Luxembourg, Esch-sur-Alzette, Luxembourg

[3] Environmental Metagenomics, Research Center One Health Ruhr of the University Alliance Ruhr, Faculty of Chemistry, University of Duisburg-Essen, Essen, Germany.

[4] AkerBP, 1366 Lysaker, Norway

[*] https://www.prospectomics.eu/

[+] A full list of authors and their affiliations appears at the end of the paper.

*Corresponding author*

Jens Kallmeyer, GFZ German Research Centre for Geosciences, Section Geomicrobiology, Telegrafenberg,

20       14473 Potsdam, Germany, email: kallm@gfz-potsdam.de

**Abstract.** All hydrocarbon (HC) reservoirs leak to some extent. When small quantities of HCs escape offshore reservoirs and migrate through overlying organic-poor marine sediments towards the surface, these HCs are often completely metabolized by microbial activity before reaching the sediment-water
25       interface. However, inconspicuous HC fluxes still affect the geochemistry of the surrounding sediment, thereby exerting a subtle influence on the composition and activity of microbial populations in shallow subseafloor environments.

In this study, we investigated how localized HC seepage affects microbial sulfate reduction in organic-poor sediment from the SW Barents Sea. We focused on three areas overlying known HC deposits and two
30       reference areas of pristine seabed for comparison. The analysis of 50 gravity cores revealed significant



variability in the predicted depth of sulfate depletion across sampling sites, ranging from 3 to 12 m below the seafloor. Although we observed nearly linear pore water sulfate and alkalinity profiles, we measured and modeled low rates (pmol $\times$ cm$^3$ $\times$ d$^{-1}$) of sulfate reduction. Metagenomic and metatranscriptomic data on functional marker genes supported microbial turnover associated with active processes of sulfate

reduction and anaerobic oxidation of methane (AOM). Marker genes for taxonomy (i.e. SSU rRNA, *rpoD*), sulfate reduction (i.e. *dsrAB*, *aprAB*), methanogenesis and methanotrophy (i.e. *mcrA*) revealed metabolic activities by a consortium of sulfate-reducing bacteria and ANME archaea, capable of harnessing energy for cell division (i.e. *ftsAZ*) from HC traces diffusing through the sediment.

Overall, our study demonstrates that the gradient in pore water geochemistry, the rates of sulfate reduction

processes, and the genetic features of microbial populations actively involved in sulfate-driven AOM processes are all affected by inconspicuous HC seepage. This slight HC seepage resulted in sulfate depletion at shallower depth and produced concomitant biogeochemical signatures in the shallow subsurface that enable the inference of deeply buried reservoirs.

## 45   1. Introduction

All hydrocarbon (HC) reservoirs leak to some degree (Hunt, 1995; Yergin, 2009; Heggland, 1998), with over 80 % of all seeps occurring directly above the reservoirs (Ciotoli et al., 2020). While large leakages result in conspicuous manifestations at the seafloor, (e.g. Cramm et al. (2021)), minor seeps usually remain invisible at the sediment-water interface (SWI). Still, sediment geochemistry and microbiology in the

vicinity of inconspicuous seepage sites may be altered by HC fluxes (Rasheed et al., 2013; Abrams, 2020). Slight geochemical changes are common in the direct surroundings of HC reservoirs and overlying sediments, yet distal manifestations also occur. As seepage propagates further upwards, traces of HCs sometimes reach the seabed (Abrams, 2020; Joye, 2020), but are often overlooked.

HC seepage triggers both geological and microbiological processes, changing physicochemical properties,

such as sediment porosity, mineral and pore water composition, as well as microbial community composition and activity (Abrams, 2020; Hvoslef et al., 1996; Joye, 2020). Likewise, methane-containing pore fluids alter the sediment porosity and compressibility (Jang et al., 2018) that are traceable in the acquisition of seismic profiles (Rovere et al., 2020), and potentially result in changes in the sediment geochemical properties (e.g. clay texture, organic matter (OM) reactivity, metal content (Chen et al.,



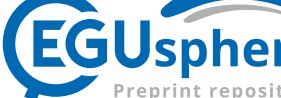

2023)). HC influx will also alter pore water geochemistry and pH, promoting ion exchange reactions, dissolution, precipitation and structural conversion of minerals (Jiang, 2012). As a consequence, specific trace metals and metalloids (e.g. Cr, Cu, As, Se, Sb) that can be metabolized by specific microorganisms (Raab and Feldmann, 2003) tend to become enriched in the sediment pore water at seepage sites (Rasheed et al., 2013; Guseva et al., 2021). Finally, most offshore sediments are composed of pelagic deposits, i.e.

clays, silts or biogenic oozes, containing only low amounts of bioavailable OM that becomes increasingly recalcitrant with depth of burial (Middelburg, 2018). Thus, the influx of HCs in such sedimentary systems is an additional source of electron donors, providing chemical energy for microorganisms and potentially fueling various metabolic processes. The metabolic potential necessary to break down and assimilate HCs has been identified in a large number of microorganisms, e.g. reviewed in Joye (2020), whose activities

may induce concomitant effects in the pore water, organic and mineral fraction of the sediment (Kim et al., 2004).

The quantitatively most important anaerobic OM degradation process in marine sediment is sulfate reduction (Jørgensen, 1982). Due to its abundance in seawater, sulfate diffuses into the sediment pore water where it is used as an electron acceptor by sulfate-reducing microorganisms (Widdel et al., 2010). Sulfate

reduction (SR) can be divided into two main pathways, namely organoclastic and methanotrophic SR. The respective overall reactions are:

Organoclastic SR:        $2\ CH_2O + SO_4^{2-} \rightarrow$        $2\ HCO_3^- + H_2S$    (Equation 1)

Methanotrophic SR:      $CH_4 + SO_4^{2-}$       $\rightarrow$       $HS^- + HCO_3^- + H_2O$        (Equation 2)

Due to the abundance of easily biodegradable OM and sulfate close to the sediment-water interface,

organoclastic SR normally prevails in anaerobic OM degradation within the uppermost part of the sediment column, leading to a decrease in sulfate concentration. Methanotrophic SR becomes predominant only when biogenic or thermogenic methane diffusing from deeper sediments reaches the zone where sulfate is still available but approaches depletion (Martens and Berner, 1974). The narrow zone where both methane and sulfate are available is referred to as the sulfate-methane transition zone

(SMTZ). Within the SMTZ, mainly methane, and potentially also other OM compounds (Beulig et al., 2019; Jørgensen et al., 2019b), are oxidized by microorganisms while sulfate is reduced (Equation 2).

Thus, the respective methane and sulfate fluxes exert control on the depth of the SMTZ (Borowski et al., 1996, 1999; Jørgensen and Kasten, 2006), which, under steady-state conditions, is usually a reflection of methanogenesis rates (Henrichs and Reeburgh, 1987) or fluxes of thermogenic methane (Hu et al., 2017).

However, the SMTZ is highly dynamic and responds relatively rapidly (i.e. within decades) to shifts in



pore water geochemistry (Sultan et al., 2016; Hong et al., 2016). Thus, microbially active subseafloor sediments exhibit significant variability in the depth of the SMTZ across different oceans, from 1 to 10 of meters on the continental shelf (Egger et al., 2018) down to ca. 100 m along the continental slope (Nunoura et al., 2009).

The subseafloor biosphere hosts phylogenetically diverse and metabolically active microbial communities (Parkes et al., 2014; D'hondt et al., 2004). Their metabolic activity depends mainly on biogeochemical cycling of nitrogen, phosphate and sulfur. At HC seepage sites, microbial communities are well-defined in terms of taxonomy and highly specialized in terms of metabolic functions, and are referred to as "seep biomes" (Ruff et al., 2015). These biomes show spatial heterogeneity, but are distinct from non-seep

environments (Pop Ristova et al., 2015). The reported key microbial guilds include methane and oil oxidizers, sulfate reducers, and nitrogen fixers. Sulfate reducers potentially found near HC seeps involve members of the bacterial phyla *Desulfobacterota* and *Chloroflexota* that can respectively use simple and halogenated HCs as electron acceptors (Kleindienst et al., 2014; Zhang et al., 2023), whereas archaeal groups, like the ANME or Asgard archaea, consume methane or make use of HCs or sulfur compounds

(Firrincieli et al., 2021; Macleod et al., 2019).

Our study aims to determine in what ways inconspicuous HC seepage modifies the diversity and activity of anaerobic methanotrophs and sulfate-reducing bacteria in surface sediments of the SW Barents Sea. We hypothesize that minor HC seepage influences pore water geochemical properties, even in near-surface sediment, and that resident microbial communities respond quickly in terms of composition and activity

to the geochemical alteration of their habitats.

For this, we collected 40 gravity cores from areas affected by light, inconspicuous HC seepage from underlying HC reservoirs (i.e. HC affected sites) and 10 gravity cores from areas without seepage from reservoirs (i.e. reference sites). We combine detailed analyses of pore water sulfate and alkalinity with potential and modeled sulfate reduction rates, dissolved methane ($CH_4$) and carbon dioxide ($CO_2$)

concentrations, and supplement them with metagenomic (i.e. total DNA) and metatranscriptomic (i.e. total RNA) data of selected samples on taxonomic and functional marker genes in order to trace metabolic activity by microbial sulfate-reducing consortia in their geochemical context.



## 2. Material and methods

### 2.1 Geological context of the Barents Sea, sampling sites

The Barents Sea is an epicontinental shelf sea surrounded by the marginal Norwegian Sea to the southwest, the Arctic Ocean to the north and the Russian island Novaya Zemlya to the east (Fig. 1a). Since its formation during the Caledonian orogeny (i.e. 490-390 Ma), the Barents Sea Basin experienced several phases of uplift and subsidence, with subsequent tilting and erosion. These geologic events resulted in the characteristic structure of the Barents Sea in the form of a succession of structural highs and basins

(Gabrielsen et al., 1990; Larssen et al., 2002).

Hydrocarbon exploration in the Barents Sea has a history stretching back several decades (Doré, 1995; Johansen et al., 1993). The sediments have been characterized in numerous studies (e.g. Sættem et al. (1991); Elverhøi and Solheim (1983)) and are mainly composed of organic-poor silty clays (Knies and Martinez, 2009; Nickel et al., 2013) originating from eroded sedimentary and igneous rocks.

During an expedition to the southern Loppa High region of the Barents Sea (Fig. 1a) from October 29[th] to November 8[th] 2021, we collected 50 gravity cores (up to 3 m in length) sediment at about 350 m water depth at three areas with known underlying HC reservoirs (i.e. Zone 1, Zone 10, Zone 12) and at two reference areas (i.e. Ref 5, Ref 6) without underlying HC reservoirs (Fig. 1b), labelled thereafter HC positive and reference sites, respectively. The cores were subsampled for sediment and pore water on

board the research vessel.

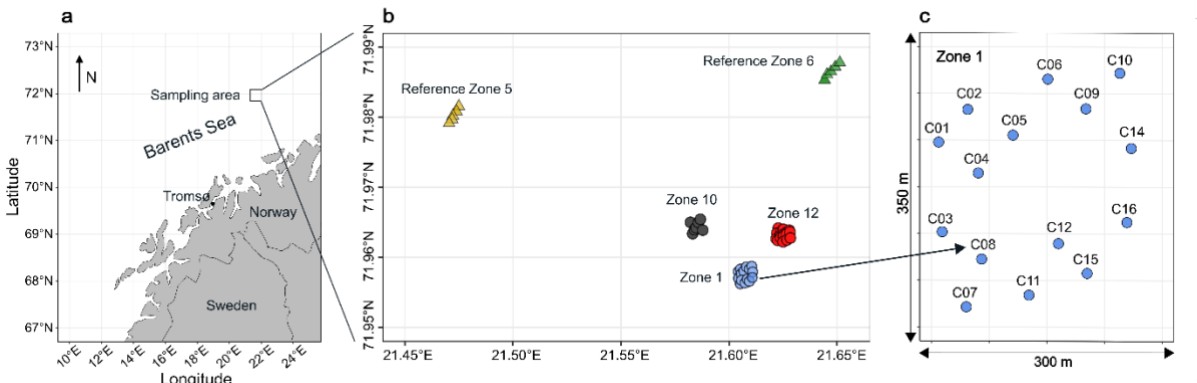

**Figure 1. Sampling locations. (a)** Map of the Norwegian coast. The square marks the sampling area. **(b)** Fifty cores were retrieved from three HC positive zones (Zone 1, Zone 10, Zone 12) and two reference



zones (Ref 5, Ref 6). **(c)** Each zone was sampled in a grid of cores, here we show Zone 1 as an example. The missing samples in the center were due to hardgrounds, preventing the deployment of a gravity corer.

## 2.2 Pore water and sediment sampling

Pore water samples were extracted from all cores at a resolution of 10 cm, using rhizons (Rhizosphere
Research Products) (Seeberg-Elverfeldt et al., 2005), collecting a minimum volume of ca. 5 mL over 24 h. The pore water was filtered through 0.2 µm pore size cellulose esters (MCE) membrane syringe filters (Merck MF-Millipore) and aliquoted as follows: 1.5 mL transferred into a plastic screw cap vial without further treatment for anion and cation measurements; 2 mL into a glass vial amended with 50 µL saturated $HgCl_2$ solution to inhibit microbial activity and sealed without headspace for alkalinity measurements
(Edenborn et al., 1985); and 1.5 mL into a plastic screw cap vial mixed with 200 µL $ZnCl_2$ (20 % weight $\times$ vol$^{-1}$) to precipitate dissolved hydrogen sulfide as ZnS for subsequent sulfide measurements. All samples were stored at +4 °C until analysis in the home lab.

Immediately after core recovery, the lowermost 20 cm of each sediment core were cut off and subsampled for molecular analyses. The sediment was pushed out of the liner and the outer 1-2 cm of sediment rims
were removed with a sterile spatula. Aliquots of the remaining sediment were transferred into gas-tight foil bags, flushed with nitrogen gas, heat-sealed and stored at -80 °C for later extraction of various biomolecules.

## 2.3 Pore water sulfate, sulfide and alkalinity

Anion concentrations were determined using an Ion Chromatography (IC) System equipped with a SykroGel
A $\times$ 300 AB-A01 column (all Sykam). The eluent contained 7.3 mg $\times$ L$^{-1}$ NaSCN and 636 mg $\times$ L$^{-1}$ $NaCO_3$. The pump rate was set to 1 mL $\times$ min$^{-1}$ and the injected sample volume was 50 µL. All samples were measured in triplicates and the results were averaged. Because of their high salt concentrations, all samples had to be diluted 1:50 with MilliQ water before injection. The detection limit of the IC system is 0.5 mg $\times$ L$^{-1}$. The average standard deviation of replicate measurements was always better than 3 %.
Sulfide concentrations were measured according to the protocol of (Cline, 1969). Aliquots with absorption values greater than 1 were diluted and remeasured. Sulfide concentrations were calculated by comparing adsorption values measured for pore water samples to those of a $Na_2S$ standard. The detection limit of the



method is 0.1 µM. All samples were measured in triplicates and the results were averaged, with standard deviation ≤ 3.5 %.

Pore water alkalinity was determined using the Visocolor HE alkalinity AL 7 kit (Macherey-Nagel GmbH). As we only had a total of ca. 2 mL of pore water per sample, we scaled down the volume of sample to 500 µL in order to perform all measurements in triplicates. For this, 10 µL of indicator was added to 500 µL of pore water and titrated with the Visocolor solution until we observed a change in color from blue to orange. Alkalinity of the pore water samples was calculated based on the titration solution volume.

Replicates differed by less than 3 % on average.

## 2.4 Sulfate reduction rates, modeled sulfate reduction rates

For sulfate reduction measurements, each gravity core was subsampled in triplicate every 40-50 cm and at the bottom end. Structurally intact sediment plugs (volume ca. 3.5 cm$^3$) were recovered by inserting 5 mL glass barrels fitted with a syringe plunger into the sediment. Following the whole core injection method

(Jørgensen, 1978), 15 µL of radioactive $^{35}SO_4$-sulfate tracer (200 kBq) were injected into the glass barrels containing the samples retrieved on the same day (Fossing et al., 2000). The glass barrels were closed with butyl rubber stoppers, and the samples incubated at *in situ* temperature (ca. 4 °C) for 24 hours. Each sample was then transferred into a previously weighed 50 mL centrifuge tube containing 10 mL of 20 % zinc acetate solution to stop all microbial activity and to precipitate all volatile hydrogen sulfide as zinc

sulfide. Each vial was then thoroughly shaken to break down all sediment aggregates and stored frozen until analysis. Blank samples were prepared by adding the radiolabeled sulfate tracer just before transferring the sample into the zinc acetate solution. The tracer blank was prepared in the same way without sediment.

Sulfate reduction rates were quantified in the home lab, using single-step cold chromium distillation

(Kallmeyer et al., 2004). The sample vials were thawed, centrifuged and the supernatant carefully removed. A small aliquot of supernatant was kept for quantification of total radioactivity, the rest was discarded. The sediment pellet was then quantitatively transferred into a distillation flask. Due to the clayey consistence of the sediments, we had to re-freeze the vials and transfer the frozen sediment pellet from the centrifuge tube to the distillation flask. Sulfate reduction rates (SRR) were calculated according

to the following formula:

Sulfate reduction rate: $$SRR = \frac{SO_4 * \varphi * a_{tris} * 1.06 * 10^6}{a_{tot} * t}$$ (Equation 3)





SRR: sulfate reduction rate [pmol cm$^{-3}$ × day$^{-1}$]; SO$_4$: sulfate concentration in the pore water [mmol × L$^{-1}$]; $\varphi$: porosity [mL × cm$^{-3}$] of the sediment, set to 0.7; a$_{tris}$: radioactivity of total reduced inorganic sulfur [cpm]; a$_{tot}$: total radioactivity used in the spiking [cpm]; t: incubation time [d]; 1.06: correction factor for isotopic fractionation; 10$^6$: unit conversion factor from [μmol × L$^{-1}$] to [pmol x L$^{-1}$].

Assuming steady state conditions, the net rate of sulfate production, or consumption, can be modeled based on the measured sulfate concentration profiles by the diffusion-reaction modeling software PROFILE (Berg et al., 1998). This software considers three different kinds of vertical transport (i.e. diffusion, bioturbation, irrigation) together with the flux across the SWI. The final output corresponds to the minimum number of intervals, as defined by their respective sulfate production or consumption rates, that are necessary to reproduce the measured pore water profiles. For the modeling calculation, we used: coefficient of sulfate diffusion in water at 4 °C D$_{Sulfate}$= 0.56 × 10$^{-5}$ [cm$^2$ × s$^{-1}$] (Iversen and Jørgensen (1993); porosity $\varphi$ = 0.7; and sulfate seawater concentration 28 mmol × cm$^3$. Because the sediment appeared to be free of macrobenthos, the biodiffusivity and irrigation coefficients were set to D$_B$= 0 cm$^2$ × s$^{-1}$, and α = 0, respectively.

## 2.5 Methane and carbon dioxide concentrations

For dissolved CH$_4$ and CO$_2$ concentrations, we subsampled the lowermost 20 cm of sediment from each gravity core directly after recovery. Using 5 mL cut-off syringes, we extracted ca. 4 cm$^3$ of sediment per core. The first and last 0.5 cm$^3$ of sediment in the syringe were discarded while the central 3 cm$^3$ were transferred into a 10 mL glass crimp vial containing a saturated NaCl solution. The vials were immediately sealed with thick butyl rubber stoppers, crimped, and stored upside down without headspace until analysis in the home lab.

Prior to measurement, we introduced 3 mL of ultrapure helium gas as headspace while withdrawing the same amount of NaCl solution from the vial. To equilibrate dissolved gases with the headspace, the content of the vials was mixed at 220 rpm on an orbital shaker for 18 h, and further vortexed to break up the remaining small clayey aggregates. We extracted ca. 350 μL of headspace from the vial using a gas-tight syringe. Prior to measurement, ca. 100 μL of the sampled gas were flushed through the injection needle, and 250 μL of gas sample were then injected into a 7890A Gas Chromatography System equipped with a flame ionization detector (FID), a thermal conductivity detector (TCD) and HP PLOT Q column (all Agilent). Oven temperature was set to 50°C, flow rate to 17.2 mL × min$^{-1}$ and pressure to 13 psi. The detectors both worked at 200°C with flow rates of 40 mL × min$^{-1}$ (FID) and 15 mL × min$^{-1}$ (TCD). The



system was calibrated, using 250 µL of analytic pure standards, injecting $CO_2$ concentrations of 310 ppm and 5270 ppm, and $CH_4$ concentrations of 10 ppm and 5170 ppm. The initial $CH_4$ and $CO_2$ concentrations were converted from ppm to molar concentrations by applying the ideal gas law.

**2.6 DNA/RNA extractions, sequencing libraries, functional marker genes**

DNA was extracted for a total of 12 samples from 8 different cores corresponding to 4, 2 and 2 samples from HC positive Zone 1, Zone 10 and Zone 12, respectively, plus 3 and 1 samples from Ref 5 and 6**.** Samples were subjected to DNA extraction using the DNEasy PowerMax Soil Kit (Qiagen, Germany) following manufacturer's instructions with few adaptations as follow. For 2.5 g of starting sediment material, the

final elution was performed in two steps using two times 2.5 ml of solution C6, two incubations and two centrifugations. Eluted DNA was then concentrated by adding 510µl of a 3M Sodium Acetate (pH 5.2) / Glycogen (0.4 µg/µL) solution and incubated for one hour at -20°C and then centrifuged to pellet DNA. After ethanol cleaning and air-drying, DNA was resuspended in 100 µL of 10mM Tris buffer. Finally, DNA was cleaned and concentrated using the Zymo DNA Clean & Concentrator (Zymo, UK) following

the manufacturer's instructions and with a final elution of 35 µL.

DNA sequencing was done by using 32.5 µL of purified and concentrated DNA for metagenomic library preparation using the QIAseq FX DNA library UDI A/B kit (Qiagen, Germany) irrespective of the individual sample concentrations. The genomic DNA was enzymatically fragmented for 10 min and DNA libraries were prepared with 7 PCR cycles for the PCR-less samples, and without dedicated PCR step for

the PCR-free samples. The average library sizes were ±400 bp. Prepared libraries were quantified using a Qubit fluorometer (ThermoFischer, USA) and quality-checked on a Bioanalyzer (Agilent, USA). Sequencing was performed on an Illumina NextSeq2000 instrument using $2 \times 151$ bp read length at the Luxembourg Centre for Systems Biomedicine (LCSB) Sequencing Platform, aiming at an average of 10 Giga base pairs (Gbps) per sequencing library.

RNA was extracted from 2x5 g of sediment per sample, using the RNeasy PowerSoil Total RNA kit (QIAGEN) according to the manufacturer's instructions, yielding 30 µL of RNA-containing extract, and quantified on a Qubit4 fluorometer using the RNA high sensitivity kit (ThermoFisher Scientific). RNA extracts were directly subjected to library preparation in technical duplicates, using the Revelo$^{TM}$ RNA-Seq High Sensitivity kit (Tecan Life Sciences) with 8 µL of RNA extractss as template. This trio kit

includes first and second strand cDNA synthesis, adaptor ligation and library amplification through 10 PCR cycles. Prepared libraries were quantified using a Qubit4 fluorometer (ThermoFisher Scientific) and





quality-checked on a Bioanalyzer (Agilent) to determine the molarity and overall size distribution of RNA molecule fragments. The average library fragment size was ± 300 bps and the yields were between 4 and 12 ng × µL⁻¹. RNA libraries displaying high and low molarities were pooled into two different batches,

and sequenced at the EMBL GeneCore (https://www.embl.org/) on a NextSeq 2000 with a P3 kit and an Illumina micro kit (both 2 × 150 bps), respectively.

*De novo* assembly of metagenomic reads was performed (Table S1) using a customized Snakemake workflow (Mölder et al., 2021), including quality-control, binning into contigs, gene prediction and annotation as described in (Bornemann et al., 2023) . Paired-end reads were interleaved, Illumina adapters and controls

removed, reads quality-based trimmed, and de-interleaved into forward and reverse read files. Bacterial and archaeal scaffolds were concatenated and mapped back to raw reads to retrieve coverage, % GC and contig lengths. Genes were then predicted by extracting open reading frames (ORFs) from metagenome-assembled scaffolds using Prodigal (Hyatt et al., 2010). Metatranscriptomic reads were quality-filtered and trimmed using Trimmomatic (Bolger et al., 2014). Functional profiles were obtained by mapping the

filtered and trimmed reads to their corresponding metagenomic databases using BWA-mem (Li and Durbin, 2009). ORFs were retrieved using Bakta (Schwengers et al., 2021) and combined with the mapping using the featureCounts tool (Liao et al., 2014).

Taxonomic identifications of all ORFs were further integrated with functional annotations by performing BlastP searches against an aggregated database of predicted proteins

(https://github.com/williamorsi/MetaProt-database), using DIAMOND protein aligner v. 0.9.24 (Buchfink et al., 2015). Cut-off values for assigning the best match to specific taxa present in the aggregated database were performed at a minimum bit score of 50, minimum amino acid similarity of 60, and an alignment length of 50 residues. We used this approach to draw conclusions about metabolic traits derived specifically from –phlyum to class taxonomic levels only (Vuillemin et al., 2020a; Vuillemin et

al., 2020b).

Here, we focus on ORFs encoding genes involved in sulfur cycling and methane-related processes only. To trace metabolic potential in anaerobic respiration of sulfur compounds (i.e. sulfate, sulfite, thiosulfate, sulfur, polysulfide), we looked for the presence of ORF- encoding genes with similarity to dissimilatory sulfite reductase (*dsr*), adenylylsulfate reductase (*apr*), anaerobic sulfite reductase (*asr*),

thiosulfate/polysulfide reductase (*phs*/*psr*), and sulfhydrogenase (*hyd*) (Vuillemin et al., 2022). For methane production and anaerobic consumption, we targeted ORFs encoding proteins of the methyl coenzyme M reductase (*mcr*). To identify and characterize metabolically active taxa, we targeted the




small subunit ribosomal ribonucleic acid (SSU rRNA) as a taxonomic marker and the filamenting temperature-sensitive mutant A-Z (*FtsAZ*) for cell division.

To confirm taxonomic assignments and metabolic activity of taxa involved in sulfate reduction and AOM processes, we performed a phylogenetic analysis of the RNA polymerase sigma factor (*RpoD*) gene proteins, *dsrA-G*, *aprAB* and *mcrA-G* that we could detect in the transcriptomes (Vuillemin et al., 2020a). All ORFs annotated to these genes were aligned against their top two BLASTp hits in the NCBI database, using MUSCLE (Edgar, 2004). Conserved regions of the alignments were selected using Gblocks 0.91b

(Castresana, 2000) with the following settings: allowing for smaller final blocks, gap positions within the final blocks and less strict flanking positions (http://phylogeny.lirmm.fr/). Phylogenetic analysis of the resulting amino acid alignments of the predicted proteins were conducted in SeaView version 5.0.5 (Gouy et al., 2010), using PhyLM maximum likelihood (Guindon et al., 2010), with BLOSUM62 as the evolutionary model and 100 bootstrap replicates (Supplementary Fig. S4-7).

**3. Results**

**3.1 Pore water geochemical profiles**

All sulfate concentrations in the uppermost profiles roughly match the sulfate concentration of the Barents Sea bottom waters (ca. 28 mM). In all cores, sulfate concentrations decrease almost linearly with depth, linear regressions showing correlation coefficients of $R^2 > 0.85$ (Fig. 2 and Supplementary Fig. S1). The

slopes of the regression lines can be divided into two groups corresponding, with one group encompassing zones Ref 5 and Ref 6, and the other group the three HC positive Zone 1, Zone 10 and Zone 12. Those groups are statistically not significant different, but the trend is clearly visible (Fig 2). The sulfate profiles from HC positive sites show a steeper decrease with depth than those from reference sites. On average, sulfate concentrations decrease by 10 mmol $\times$ m$^{-1}$, which would correspond to complete sulfate depletion

at roughly 3 m below seafloor (mbsf) at HC positive sites. At reference sites, the average slope is 3 mmol $\times$ m$^{-1}$ with little variations, which implies that sulfate would be depleted at ca. 12 mbsf.





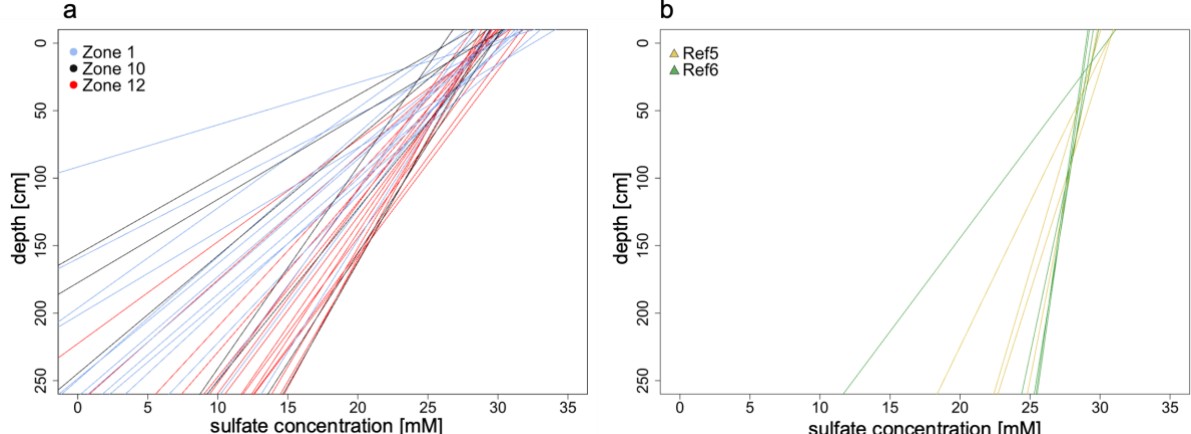

**Figure 2. Regression lines for all sulfate pore water concentration profiles.** All profiles are linear
($R^2 > 0.85$) and show a decreasing trend with depth. The regression lines calculated for **(a)** the HC positive
sites (Zone 1, Zone 10, Zone 12) show a steeper slope than those calculated for **(b)** reference sites (Ref 5,
Ref 6).

All alkalinity profiles display a clear linear increase with depth ($R^2 \geq 0.85$). From 5 mmol $\times$ L$^{-1}$ at the SWI,
pore water alkalinity increases differently with depth, corresponding to 8, 5 and 6 mmol $\times$ L$^{-1}$ $\times$ m$^{-1}$ at HC
positive Zone 1, Zone 10, and Zone 12, and $\leq 1$ mmol $\times$ L$^{-1}$ $\times$ m$^{-1}$ at Ref 5 and Ref 6, respectively (Fig. 3
and Supplementary Fig. S2).

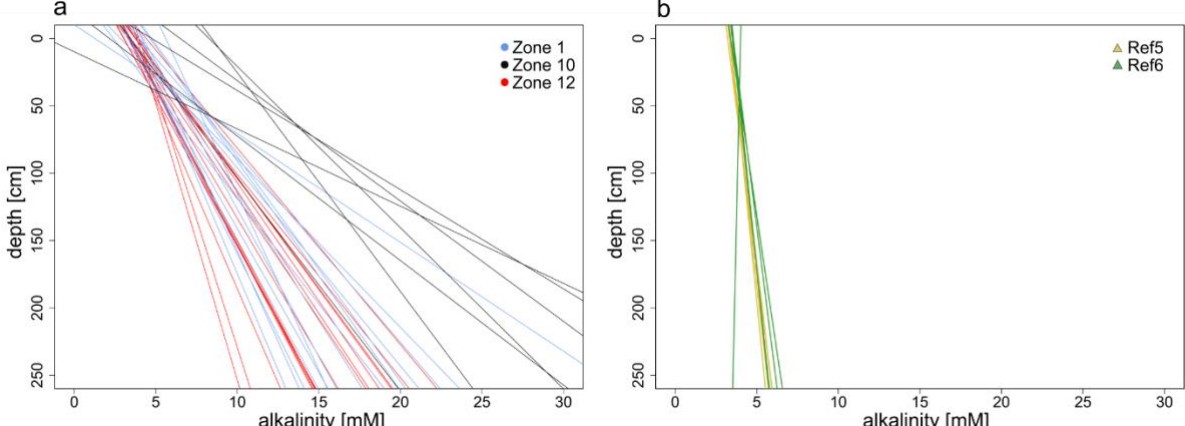



**Figure 3. Regression lines of pore water alkalinity.** All profiles are linear ($R^2 \geq 0.85$) and show an increasing trend with depth. The regression lines calculated for the HC positive zones (**a**) show a less steep gradient than those of the reference zones (**b**).

Based on the slopes of pore water profiles, the calculated alkalinity fluxes are larger at HC positive sites than at reference sites (Fig. S3). The same applies to sulfate fluxes, but differences across HC positive and reference sites are less pronounced (Fig. S3). In comparison, variations observed within each sampling site are small.

Sulfide could hardly be detected in pore water samples collected from reference sites with most values below 1 µM. In contrast, sulfide was detected in most of the cores from HC positive sites. Especially at Zone 1 and Zone 10, sulfide concentrations increase linearly with depth (Fig. 4). The majority (ca. 70 %) of the cores from Zone 1 show an increase of 2.5 mM $\times$ m$^{-1}$ in sulfide concentrations. One core (Z01C03) stands out, reaching sulfide concentrations of ca. 11 mM at 1 mbsf. Several cores (ca. 57 %) from Zone 10 show a binary distribution of sulfide concentrations, corresponding either to linear profiles reaching ca. 1mM sulfide at 1 mbsf, or to a steeper increase reaching ca. 3.7 mM at the same depth. At Zone 12, sulfide is detectable in only 2 of the 17 cores, with concentrations that increase much less compared to Zone 1 and

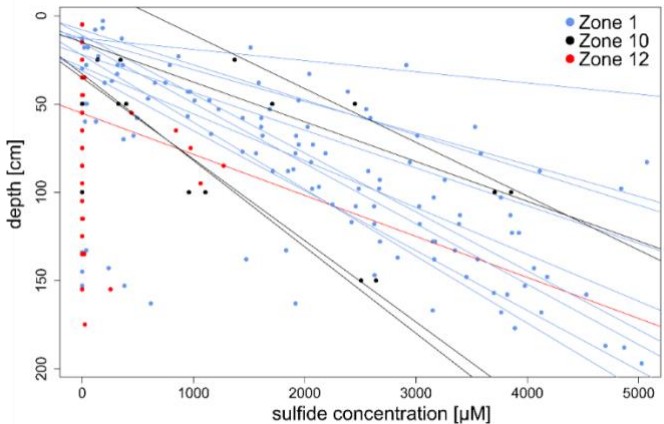

Zone 10.

**Figure 4. Sulfide concentrations measured for all cores from HC positive sites (dots) and their corresponding regression lines when sulfide concentrations were quantifiable.** The majority of the cores from Zone 1, several cores frome zone 10 show an increase in sulfide concentration with depth. In zone 12 sulfide is hardly detectable and only one core shows clear increase with depth.



### 3.2 Measured and modeled sulfate reduction rates

SRR were detectable in only 29 out of the 50 cores (Fig. 5), with rates between 4 and 320 pmol $\times$ cm$^{-3}$ $\times$ d$^{-1}$. In 403 of the 455 samples across the 50 cores analyzed (i.e. 89 %), SRR were below the detection limit (1 pmol $\times$ cm$^{-1}$ $\times$ d$^{-1}$). The occurrence of very low but detectable SRR increased with sediment depth, 350 which mostly stands true for HC positive sites (Fig. 5).

Diffusion-reaction modeling with PROFILE (Berg et al., 1998) suggests steady SRR in 47 of the 50 cores (Supplementary Fig. S1), implying that SRRs remain constant over the entire length of the cores. The modeled SRR profiles show SR activity (i.e. > 0 pmol $\times$ cm$^{-3}$ $\times$ d$^{-1}$) in 33 cores, of which 6 display rates $\leq 0.01$ pmol $\times$ cm$^{-3}$ $\times$ d$^{-1}$, and 27 with supposed SRR $\geq 1$ pmol $\times$ cm$^{-3}$ $\times$ d$^{-1}$. For 3 specific cores (i.e. 355 Z01C02, Z01C09, Z12C17), PROFILE proposes two intervals with different SRRs, suggesting that SR processes may only be active in the upper 50 cmbsf and inactive in the sediment below.

The modeled and measured SRR are indicative of net and gross turnover rates, respectively. The gross rate reflects microbial turnover whereas the net turnover rate includes the recycling of reduced sulfur compounds (Berg et al., 1998). The modeled (i.e. net) rates for most samples are higher than the measured 360 (i.e. gross) rates, which is rather unusual and can be explained by very low values whose absolute changes have a drastic effect on the calculated values. The modeled and measured SRR are nevertheless within



the        same        order        of        magnitude,        and        thus        comparable.

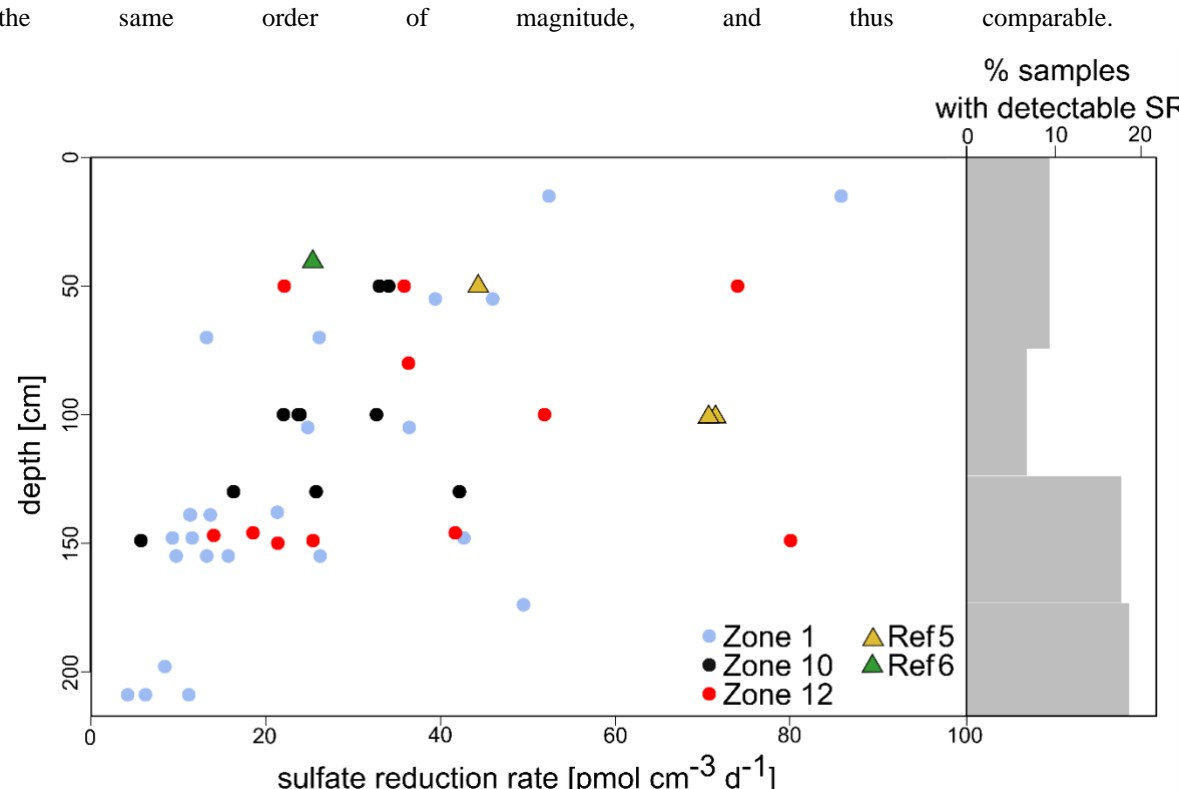

**Figure 5. Sulfate reduction rates (SRR) and percentages of samples with sulfate reduction rates above the detection limit.** Low SRR became increasingly detectable with increasing sediment depth. The number of samples with measurable SRR (% grey bars) is higher in sediments from HC positive sites (dots) compared to reference sites (triangles). n.B. due to a lack of samples, the two uppermost intervals were merged.

### 3.3 Methane and carbon dioxide concentrations

All samples contain measurable methane concentrations, ranging from 0.08 to 19.79 $\mu$mol $\times$ L$^{-1}$. All samples from reference sites contain less than 5 $\mu$mol $\times$ L$^{-1}$ of methane, only in some samples from HC positive sites and only below 100 cm sediment depth (Fig. 6) we were able to detect any higher concentrations. $CO_2$ was detectable in all samples, with concentrations ranging from 29 $\mu$mol $\times$ L$^{-1}$ to 512 $\mu$mol $\times$ L$^{-1}$, but without any specific trend with depth (Fig. 6). Such low methane and carbon dioxide concentrations prevented        any        measurement        of        their        stable        isotopic        signatures.



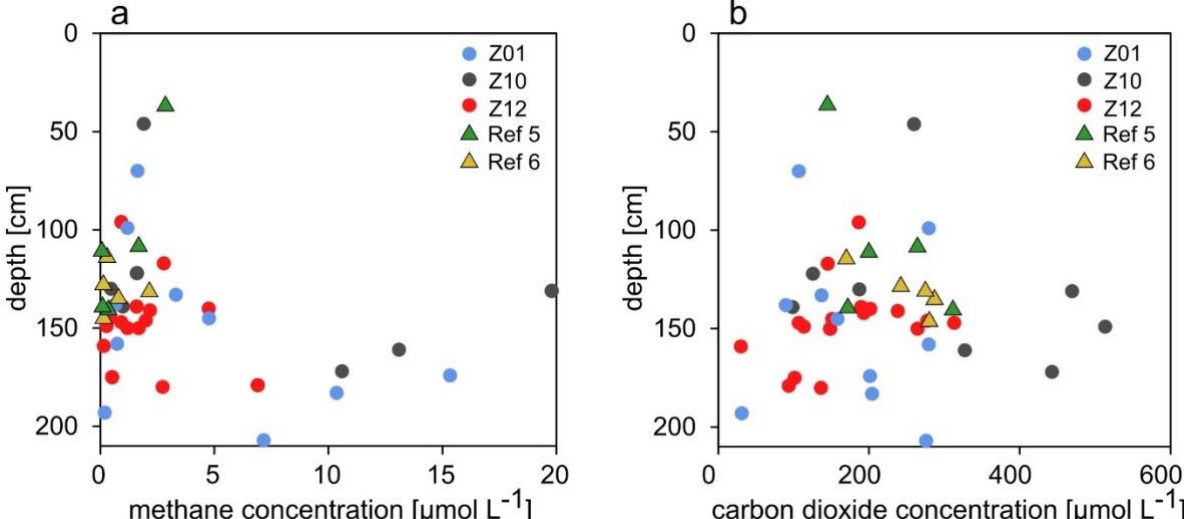

**Figure 6. Concentrations of dissolved methane and carbon dioxide.** (a) At HC positive sites methane concentrations below 100 cm depth, whereas at reference sites methane concentrations remain below 5 µmol L$^{-1}$. (b) CO$_2$ concentrations do not display any specific trend with sediment depth across samples.


### 3.4 Statistical correlations

To identify geochemical differences between HC positive and reference sites, we performed a Pearson correlation analysis. Correlation coefficients calculated for each separate gravity core (Table S2) indicate that alkalinity is strongly negatively correlated ($p \geq -0.85$) with sulfate concentrations in sediments from

HC positive sites. The same applies to quantifiable sulfate and sulfide concentrations ($p \geq -0.7$). For reference sites, alkalinity and sulfate concentrations are also strongly correlated ($p \approx 0.8$), whereas sulfide concentrations are below the detection limit and hence have been omitted for the correlation analysis.

We established ratios of sulfate to alkalinity (HCO$_3^-$) fluxes and compared them to those expected from organoclastic (1:2 based on Eq. 1) and methanotrophic (1:1 based on Eq. 2) SRR activities. At reference

sites, the average ratio of sulfate to alkalinity fluxes equals 2:1 whereas it is 1.25:1 (5:4) at HC positive sites, imply that either HCO$_3^-$ is being removed from the pore water, or that pore water sulfate is eventually





being                    replenished                    via                    sulfide                    oxidation.

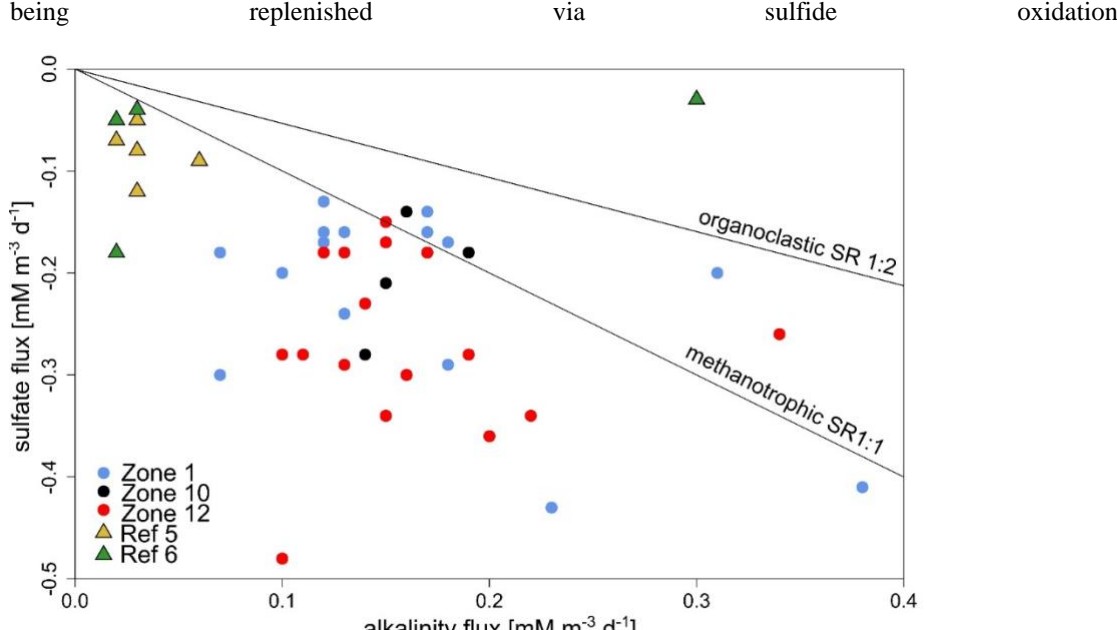

**Figure 7. Relationship between sulfate and alkalinity fluxes at HC positive and reference sites.** The
majority of samples from HC positive sites (dots) have lower alkalinity than sulfate fluxes. This is even
more pronounced at reference sites (triangles). The black lines indicate the theoretical 1:1
(methanotrophic SR) and 1:2 (organoclastic SR) ratio.

### 3.5 Functional marker genes specific to the SMTZ

From the 12 sequenced metagenomes, we extracted a total of 307,234 ORFs with multiple annotations that
corresponded to 821,850 predicted genesFrom the 5 sequenced metatranscriptomes, we extracted a total
of 31,093 ORFs corresponding to 47,316 predicted genes (Supplementary Table S2)**.**

For each of the targeted marker genes, we plotted the relative percentage of total predicted genes and their
corresponding taxonomic assignments at the phylum and class level (Fig. 8). Taxonomic assignments of
*SSU* rRNA and *FstAZ* proteins show that metabolically active and growing microbial populations HC
positive sites are affiliated with the phyla Chloroflexota, Desulfobacterota and Euryarchaeota, and more
specifically with anaerobic methanotrophs (i.e. ANMEs). The phyla Desulfobacterota and Chloroflexota
are actively engaged in processes of sulfate reduction, as shown by the expression of ORFs encoding the
*dsr* and *apr* genes (Fig. 8). ORFs encoding *hyd* genes are expressed by taxa related to the phylum




Asgardarchaeota, indicating metabolic capability to use reduced sulfur compounds (e.g. polysulfide, elemental sulfur). However, their overall expression levels appear limited and Asgardarchaeota do not seem to actively divide. Taxonomic assignments of expressed ORFs encoding *mcr* genes show an increased abundance of methanotrophic ANMEs over methanogenic Halobacterota. At reference sites, Chloroflexota appear to be the only phylum capable of growing and dividing in these organic-poor

sediments. However, concomitant expression of functional marker genes involved in sulfate reduction remains below the limit of detection. We acknowledge that the non-detection of certain genes may also result from limited sequencing depths.

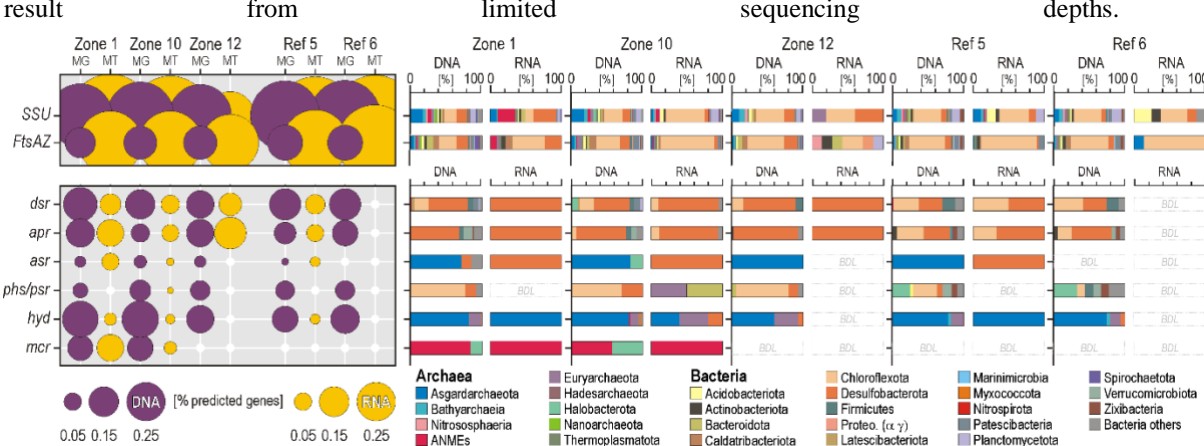

**Figure 8. Open reading frames encoding functional marker genes involved in sulfate reduction,**

**production and anaerobic consumption of methane and their taxonomic assignments at the phylum to class level.** The relative abundances of certain phyla increase at HC positive sites. Concomitant expression of *dsr* and *apr* genes by Desulfobacterota, *phs* genes by Chloroflexota, *hyd* genes by Asgardarchaeota, and *mcr* genes by ANMEs and Halobacterota are consistent with methanotrophic sulfate-reducing activities. ***Abbreviations - *** *dsr*: dissimilatory sulfite reductase; *apr*: adenylylsulfate

reductase; *asr*: anaerobic sulfite reductase; *phs/psr*: thiosulfate and polysulfide reductase; *hyd*: sulfhydrogenase; *mcr*: methyl-coenzyme M; MG: Metagenomics; MT: Metatranscriptomics; BDL: below detection limit**.**





## 4. Discussion

### 4.1 Pore water geochemical profiles correlate to meta-omics features

Pore water profiles reflect the spatial distribution of net turnover processes. The shape of the profile allows to distinguish intervals in which transport is purely diffusive (i.e. linear profile) from those in which there is either net production (i.e. convex profile) or consumption (i.e. concave profile) of chemical species (Schulz, 2006; Hesse and Schacht, 2011).

Based on the largely linear profiles for pore water alkalinity, sulfate and sulfide (Figs. 2-3, Figs. S1-S2), we

infer the absence or near-absence of net sulfate consumption in the depth interval covered by the cores, and a sink for sulfate as well as a source of sulfide and alkalinity at greater depth. Microbially-mediated AOM with sulfate as electron acceptor yields hydrogen sulfide, bicarbonate and water (Eq. 2), providing an explanation for both sink and source (Martens and Berner, 1974; Iversen and Jorgensen, 1985), assuming sufficient supply of methane at greater depth. The occurrence of AOM at deeper depth is further

supported by the relationship between sulfate and alkalinity fluxes (Fig. 7) and negligibly small gas concentrations ($CH_4 < 10 \, \mu M \, L^{-1}$) at the bottom end of all cores (HC positive sites and reference sites), indicating almost complete consumption of methane originating from biogenic or thermogenic sources below our sampling interval.

Linear concentration profiles are only obtained in regions with no net consumption or production of the

respective compound (Schulz, 2006). Despite the apparent linearity of sulfate profiles, we observed very low SRR ($\leq 300 \, pmol \times cm^{-3} \times d^{-1}$), confirming previous reports of sporadic SRR in Barents Sea sediments with similar rates above the SMTZ (Nickel et al., 2012). In our case sulfate reduction is mainly detectable at HC positive sites, with a higher occurrence in deeper sediments (Fig. 5). With seepage supplying electron donors like methane and/or other HCs from deeper reservoirs to near-surface sediment,

sulfate reduction is fuelled by those HCs rather than from the buried and more recalcitrant organic matter. Inconspicuous supplies of methane and/or other HCs to the sediment above can stimulate SRR by providing alternative electron donors. Such sporadic energy supply promotes geochemical conditions that stimulate benthic HC-degrading microbial communities. Although these communities can be physiologically and phylogenetically diverse, they are mainly composed of anaerobic guilds of

methanogens, methanotrophs, sulfate reducers and fermenters (Teske, 2019).

The analysis of functional marker genes supports the notion of sporadic HC supply, providing complementary insights into the metabolic potential and activity of specific microbial populations in the sediment. The



occurrence and abundance of *dsrAB*, *aprAB*, and *mcrA* genes supports interpretations of geochemical data in terms of sulfate-driven AOM processes and metabolic activity (Fig. 6). Taxonomic profiling of populations expressing ORFs encoding genes involved in sulfate reduction (i.e. *dsrAB*, *aprAB*) reveals Desulfobacterota and Chloroflexota as the prevalent bacterial phyla constituting the consortium of sulfate-reducing bacteria at HC positive sites (Fig. 8, Supplementary Figs. S5-S6). Further, ORFs encoding genes involved in anaerobic production and consumption of methane (i.e. *mcrA*) clearly show that the clade ANME-I exhibits concomitant metabolic activity towards anaerobic oxidation of methane (Fig. 8, Supplementary Fig. S7), namely they express ORFs encoding the reversible *mcrA* gene. The higher expression levels of methanotrophic clade ANME-I over methanogenic Halobacterota also suggest that metabolic activity by ANMEs could be sustained by an alternative source of methane (Dong et al., 2020), although cryptic production of biogenic methane by the ANME-1 clade has also been observed in experiments replicating SMTZ conditions (Beulig et al., 2019). Because methane concentrations measured at both HC positive and reference sites were similarly low, it appears unlikely that AOM processes are sustained by *in situ* production of biogenic methane only (Dong et al., 2020). Finally, the phylum Asgardarchaeota expressed ORFs encoding the *hyd* genes, and thereby appears metabolically active, using various sulfur and sulfide species, potentially produced through sulfate reduction.

In contrast, Chloroflexota was the only phylum at reference sites where its metabolic activity was adequate to achieve cell division (i.e. *FstAZ*). In the absence of a clear signal towards active sulfur turnover processes (Supplementary Fig. S5), we infer that these Chloroflexota could persist and gain energy through fermentation processes in organic-lean marine sediments instead (Vuillemin et al., 2020a).

Together, these microbial patterns extracted from meta-omics corroborate the geochemical data, arguing in favor of AOM processes concomitant to sulfate reduction at HC positive sites, and for limited *in situ* production of biogenic methane at both HC positive and reference sites. The production of reduced sulfur compounds could apparently stimulate metabolic activity by Asgardarchaeota. Due to their metabolic versatility, Chloroflexota were found to be active at both HC positive and reference sites, which rank them as poor microbial indicators for inconspicuous HC seepage (Iasakov et al., 2022).

## 4.2 Geochemical indicators for inconspicuous HC seepage

In addition to the microbial patterns extracted from meta-omics that allow to distinguish between HC positive sites and reference sites, and the occurrence of sulfate reduction in the upper sediment, some geochemical features are also useful to distinguish HC and reference sites in the Barents Sea. Sediments of the SW



Barents Sea are organic-poor (TOC <0.5 %), and thus characterized by a lack of electron donors (Knies and Martinez, 2009) and low microbial activity (Nickel et al., 2012; Nickel et al., 2013) with consequent

sulfate penetration to greater sediment depths. In organic-rich sediments, any signal inherent to minor HC seepage fueling SR would be difficult to detect as it would be covered by organoclastic SR processes. The organic-lean nature of Barents Sea sediments should enable the detection of discrete HC seeps, because the slightest supply of electron donors leads to a direct increase in the metabolic activity of HC-degrading microbial populations. This slight increase in metabolic activity can then be discriminated from

the background signal observed at reference sites. Diffusion of HCs was thus expected to promote sulfate consumption, resulting in its depletion in the pore water at shallower sediment depths. This, combined with variable fluxes of thermogenic and biogenic methane, can account for variations in the depth of the SMTZ.

At different locations of the SW Barents Sea, the depth of the SMTZ, or rather the sulfate depletion depth,

was postulated, for instance, at 37 mbsf in the Hammerfest/Loppa High and the Tromsø Basin/Ingøydjupet area (Nickel et al., 2012), and between 3.5 and 29.2 mbsf in the Ingøydjupet area (Argentino et al., 2021a). These values are consistent with the global average SMTZ depth (12.8 ± 12.1 mbsf) on the continental slope (Egger et al., 2018). For comparison, at the Vestnesa Ridge which is under the influence of gas hydrates, the depth of the SMTZ fluctuates massively between seepage (ca. 1 mbsf)

and non-seepage (ca. 3 to 75 mbsf) sites (Hong et al., 2016). Similarly, in the Barents Sea sediments with and without methane seeps show an SMTZ located between a few centimeters depth and about 2 mbsf (Argentino et al., 2021b).

Linear extrapolation of sulfate profiles (Fig. 5) indicates complete sulfate depletion, indicative of an SMTZ, at 12 and 3 mbsf for the reference and HC positive sites, respectively. We suggest that the difference in

sulfate penetration depth is caused by upwards migrating HCs.

The sulfate and alkalinity fluxes inherent to organoclastic (Eq. 1) and methanotrophic (Eq. 2) SR have theoretical ratios of 1:2 and 1:1, respectively, while mixed SR processes have intermediate values. Current ratios of sulfate to alkalinity fluxes (i.e. 2:1 at HC positive sites, 5:4 (i.e. 1.25:1) at reference sites) imply that either $HCO_3^-$ is being removed from the pore water, or that pore water sulfate is eventually being

replenished via sulfide oxidation. While dissolved inorganic carbon (DIC) can be assimilated via acetogenesis, methanogenesis, or methanotrophy (Kellermann et al., 2012), carbonate minerals are known to precipitate from the pore water as by-products of AOM processes (Turchyn et al., 2021). Up to 80 % of the sulfide produced through SR can also be converted back into sulfate through reoxidation of reduced



sulfur compounds (Jørgensen et al., 2004) and disproportionation (Pellerin et al., 2015; Jørgensen et al., 2019a). Finally, the dissolution of sulfate minerals, such as barite ($BaSO_4$), may also act as an additional source of sulfate (Griffith and Paytan, 2012).

Altogether, this suggests either the reoxidation of sulfide, or the removal of bicarbonate from the pore water, e.g. by mineral precipitation, or both. The former is further supported by concomitant sulfate reduction, detected by radiotracer incubations, and microbial gene expression patterns. Thus, we could identify several biogeochemical features that argue for ongoing sulfate reduction (most likely AOM) at HC positive sites below our sampled depth range, which allow their discrimination from reference sites.

## 5. Conclusions

Our study, combining modeled and measured SRR as well as pore water geochemistry with meta-omics of functional marker genes, showed remarkable differences between samples originating from a pristine seafloor and those from areas affected by small and inconspicuous HC seepage. The linear shape of pore water profiles was indicative of purely diffusive transport under steady-state conditions and allowed estimation of the depth of sulfate depletion, which is markedly different between HC positive and reference sites. At the HC positive sites, we found higher pore water fluxes of sulfate, sulfide and alkalinity, and could also demonstrate active microbial SR by turnover measurements and modeling. Furthermore, meta-omics provided additional evidence for enhanced metabolic activity of microbial sulfate reduction coupled to AOM processes at HC positive sites, which could be clearly discriminated from the background signal inherent to fermentative activities and quasi-absence of *in situ* methane production by the subseafloor biosphere. These observations demonstrate that minor HC seepage, which is not even visible at the seafloor significantly influences sedimentary biogeochemical cycles driven by microbial populations resident in these organic lean sediments.

## 6. Data availability

**ORFs sequences and their BlastP outputs are available as Supplementary Information. All geochemical data are available at PANGAEA (https://www.pangaea.de/) under accession number fill.**



## 7. Author contributions

JK and RDP organized the sampling cruise. ES and JK designed the study and sampled the cores. ES performed sulfate, alkalinity, SRR and gas analyses. AV, BK, CL and AS performed DNA and RNA analyses. ES and AV wrote the manuscript and all authors contributed to writing and revision.

## 8. Competing interests

The authors declare that they have no conflict of interest.

## 9. Acknowledgments

We thank all crew members of the SV *Sverdrup*, as well as Steffen Okolski, Jan Axel Kitte and Edgar Kutschera for their great help during the sampling cruise. The help of Steffen Okolski during ion chromatography, alkalinity and sulfide measurement procedures and Simone Bernsee during radioisotope experiments is acknowledged.

## 10. Financial support

This research has received funding from the European Union's Horizon 2020 research and innovation programme under grant agreement no. 899667.

## PROSPECTOMICS Consortium

The principal investigators of the PROSPECTOMICS project are Jens Kallmeyer[1], Paul Wilmes[2], Alexander J. Probst[3], Dörte Becher[4], Thomas Rattei[5], and Rolando di Primio[6]. The project managers are Aurèle Vuillemin[1], Cédric C. Laczny[2], André R. Soares[3], and Anke Trautwein-Schult[4]. Scientists and technicians include Ellen Schnabel[1], Kai Mangelsdorf[7], Steffen Okolski[1], J. Axel Kitte[1], Benoit J. Kunath[2], Oskar Hickl[2], Tuesday Lowndes[2], Zainab Zafar[2], Sarah Esser[3], Anne Ostrzinski[4], Sebastian Grund[4], and Alexander Pfundner[5].

[1] GFZ German Research Centre for Geosciences, Section Geomicrobiology, Telegrafenberg, 14473 Potsdam, Germany; [2] Luxembourg Centre for Systems Biomedicine, University of Luxembourg, Esch-sur-Alzette; [3] Environmental Metagenomics, Research Center One Health Ruhr of the University Alliance Ruhr, Faculty of Chemistry, University of Duisburg-Essen, Essen, Germany; [4] Department of Microbial



Proteomics, University of Greifswald, Greifswald, Germany; [5]Computational Systems Biology, Centre for Microbiology and Environmental Systems Science, University of Vienna, Vienna, Austria; [6]Aker BP ASA, Sandvika, Viken, Norway; [7]GFZ German Research Centre for Geosciences, Section Organic Geochemistry, Telegrafenberg, 14473 Potsdam, Germany.

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
