# Peer review of "Influence of minor hydrocarbon seepage on sulfur cycling in marine subsurface sediments"

_EGUsphere, 2024_

## Author Comment (AC1)

**Reviewer 1**

The overall conclusions of this study – sulfate-dependent AOM by ANME archaea in the seep sediments, and slow organic matter remineralization at the background sites – are plausible, but the manuscript as a whole makes a somewhat improvised impression and many datasets are not presented to their best advantage. The metagenomic and transcriptomic analysis remains at a fairly generic level and does not comment on interesting results (for example the Chloroflexota dsr genes); the phylogenetic data are used to very limited effect. Much more is possible by going beyond phylum-level generalizations; it is very hard to say anything meaningful if the analysis and discussion remain stuck on this level. Phylogenies exist, and they should be used and explored.

To summarize, the manuscript needs more work and additional analyses to sharpen the conclusions.

*Answer: We would like to thank Prof. Teske for spending time and effort into revising our paper. We agreed to most of his suggestions to improve the paper. Here, we provide point-by-point responses to his comments.*

*Original comments by the Reviewer are listed in regular font while our answers appear in italic font.*

**Manuscript comments**

Unpleasant surprises in the Introduction:

Line 35 and ff. The presence of marker genes reveals potential, not activity. For the latter, you would need transcriptional data.

*Answer: Wording was changed.*

Lines 102 to 105: Chloroflexota are sulfate reducers? Sulfate reducers (Desulfobacterota) use simple and halogenated HCs as electron acceptors (not sulfate?). The involvement of the Asgards in methane and HC degradation is also highly debatable and very likely applies to specialized lineages only; generalizations across a phylum (or superphylum, see Eme et al. 2023 Nature) are not helpful. These lines need to be rewritten and disentangled, to avoid nonsensical statements.

*Answer: Thank you for catching our mistake. Indeed, sulfate reducers (Desulfobacterota) use simple and halogenated HCs as electron donors with sulfate as electron acceptor.*

Line 290: Did you obtain and analyze gene transcripts (mRNA)? Apparently yes, but this needs to be introduced more clearly.

*Answer: Yes, we analyzed gene transcripts and revised the text accordingly to emphasize this.*

Lines 306/7: Why are these groups not "statistically significant"? How was this tested?

*Answer: The t-test statistics were added to the text.*

Concerning figures 2 and 3, I would bring the original porewater profiles from the supplements back into the main manuscript, to demonstrate that the profiles are in fact quite linear. Or make a suitable selection from the original profiles in a nicely designed full-page figure ("Representative profiles from …") so that the reader can see them without having to check the supplements. There is no reason to hide the real data in the supplements. After all, the central argument of this manuscript depends on these profiles!

*Answer: We have combined Figures 2, 3 and 4 and added some representative profiles to show that the linear regression is justified. The new figure was implemented to the main text (Fig. 2). We do not think it to be such a good idea to simply compile representative profiles from the supplement, as we believe that the trends we report in the main text can be captured much more quickly by depicting regression lines plotted jointly instead of overwhelming the reader with a huge number of individual plots. We have also plotted the data points together with their regression lines, but the resulting plot appears too busy and one can no longer tell the individual cores apart (see plot here under).*

[Figure]

The Metagenomics section bumps into some very interesting questions, and more or less ignores them. Line 410 ff: Are the Chloroflexota dsr genes functioning in the oxidative or reductive direction? Do they function in assimilatory or dissimilatory sulfate reduction? To which group of the Chloroflexota (a huge and highly diverse phylum!) do they belong? Assimilatory sulfate reduction is documented for Chloroflexota: Zheng R, Wang C, Sun C. 2024. Deep-sea *in situ* and laboratory multi-omics provide insights into the sulfur assimilation of a deep-sea *Chloroflexota*bacterium. mBio 15:e00004-24. https://doi.org/10.1128/mbio.00004-24

*Answer: We decided to combine the phylogenetic trees plotted for aprAB, dsrAB and mcrA genes into one single Figure and implement it to the main text (Figure 7, also see here under). To address a further comment by the Reviewer, we also implemented a phylogenetic tree for hydB-G gene transcripts. The aprAB and dsrAB trees clearly show that none of these genes correspond to their reverse function. Moreover, the sediment being fully anoxic, this would be quite surprising to detect oxidation of sulfur via reverse aprAB or dsrAB (mostly known in aerobic Gammaproteobacteria) and the genes annotated as aprAB and dsrAB are predicted to function in dissimilatory sulfate reduction.*

*Concerning taxonomic assignment to putative sulfate-reducing Chloroflexota, we could reach down to the order level using the NCBI database, i.e. Dehalococcoidales. Otherwise, anaerobic oxidation of methane is actively performed by ANME-1 archaea. With regards to the hydB-G genes, we can show that the class Lokiarchaeia and Thorarchaeia express genes involved in sulfur/polysulfide reduction (i.e. sulfhydrogenase).*

[Figure]

The Figure 6. Carbon dioxide per se is not abundant in normally buffered seawater; DIC occurs mostly as bicarbonate and some carbonate anions, and also microbially produced $CO_2$ will enter the carbonate equilibrium (and thus magically disappear if you look for $CO_2$ only). This is the likely reason why the HC-rich samples and the background sediments do not show a significant difference in $CO_2$ concentrations.

*Answer: This is a good argument which has been added to the results section as there is no corresponding section in the discussion.*

Figure 7. Are you using "alkalinity flux" in the sense of total DIC flux ($CO_2$ plus bicarbonate plus carbonate), or are you thinking of seawater alkalinity (which includes buffering contributions from other seawater compounds as well)?

*Answer: We measured total alkalinity. We have added an explanatory sentence to the manuscript in chapter 3.4.*

Figure 8 is unreadable. It is impossible to tell which colors belong to archaea and to bacteria (for example, ANME has the same shade of red as Nitrospirota). Try another solution, either by providing contrasting colors to bacteria and archaea (blue, green and yellow for bacteria; red and purple for archaea), or design separate plots for bacteria and archaea.

*Answer: Taxonomic assignments to Bacteria start with Acidobacteriota (signified in bright yellow). Most of the functional genes are assigned to Desulfobacterota and Chloroflexota whose color coding does not overlap with the one used for Archaea.*

Line 470: Do you have 13C data for methane to distinguish biogenic from deeply-sourced (thermogenic) methane? Is there anything usable about Barents Sea methane in the literature?

*Answer: In principle, it is possible to distinguish isotopically between thermogenic and biogenic methane on the basis of the isotopes, but in our case this is not possible because the methane concentrations in the sediment samples were too low for isotopic measurements (see also results in chapter 3.3).*

Line 480: How does sulfide production stimulate the Asgards? This is again an example of overly generic statements that really mean nothing. Remember that the Asgards consist of multiple phylum-level lineages (Eme et al. 2023 Nature); is it likely that they will all behave in the same way when tickled by sulfide?

*Answer: We identified sulfhydrogenase (hydB and hydG), aka sulfur reductase (EC 1.12.98.4), which is typically involved in respiratory sulfur and polysulfide reduction. The reduced sulfur species targeted by this metabolic process are inferred to derive from dissimilatory sulfate reduction. The corresponding ORFs were taxonomically assigned to the phylum Asgardarchaeota, i.e. class Lokiarchaeia and Thorarchaeia. To support this statement, we provide a phylogenomic tree (see here under and Figure 7D in the main manuscript).*

[Figure]

Line 485: What are the geochemical features that can be used to tell apart seep and background sites? The discussion always returns to sulfate in its various guises (penetration depth, SMTZ …). Low organic carbon content is discussed, not as a diagnostic feature but as a factor that helps HC detection by keeping the heterotrophic background down.

*Answer: The sites were selected by our industry partner Aker BP, based on their extensive exploration activities in the area, including exploratory drilling. From the data presented in our study, none of the datasets alone can be used to differentiate between HC and Reference sites. Only through a combination of various parameters, including the geochemical data can we make a distinction.*

Line 500 ff: the enormous variability of SMTZ depth in continental margin sediments argues against using this criterion for identifying slow seep areas

*Answer: We tend to disagree with the Reviewer's comment. Although the high variability of the SMTZ at the continental margin means that the SMTZ depth cannot generally be used as a criterion for the identification of seeps, a depth shift on the local scale (our HC and Reference sites are just a few km apart) can certainly serve as a criterion. We clarified this in the text.*

Lines 510 ff: these discussion paragraphs come across as somewhat generic. For example, a more careful analysis of the metagenomic data could help to identify whether DIC removal or sulfate recycling via sulfide oxidation are more likely. For example, do you have dsr genes that function in the oxidative direction? These can be told apart from their reductive cousins (Dahl et al. 2005. J Bacteriol. 187(4):1392-404). Also, what is known about the redox status of these Barents Sea sediments? Oxygen, nitrate, metals – anything that could serve as an electron sink for sulfur oxidation?

*Answer: From previous work in the area we know that the sediment is fully anoxic below the uppermost few cm. Also, the aprAB and dsrAB trees clearly show that none of these genes correspond to their reverse function. We did not identify dsrC, which potentially works in sulfur oxidation. We assume that carbonate ions are originating from AOM at greater depths, diffusing upward and eventually precipitate as calcium carbonate. However, given the sulfate concentration profiles and the scarce methane data, it becomes clear that AOM is not a quantitatively important process in the cored depth interval. Also, the low level of detection of transcripts related to AOM suggest that the imprint on the geochemistry is minimal to non-detectable.*

*The low level of transcripts related to AOM suggest that the imprint on the geochemistry is minimal to non-detectable.*

*We are somewhat surprised that the Reviewer considers the analysis of our dataset to be quite generic since, to our knowledge, there is no other paper that presents data for such a great number of sampling sites, and such a wide range of geochemical and molecular biological parameter, with all samples being processed to identical standards, thereby allowing for unprecedented compatibility of the individual datasets. Therefore, the paper not only explains the influence of HC seepage, but also shows the spatial variability of various parameters.*

A concluding note about the phylogenetic gene trees in the supplements – they are barely mentioned in the manuscript and not really used for anything. However, they could demonstrate phylogenetic affinity for particular groups and lineages within major phyla, and thus they could sharpen the discussion beyond generic phylum-only generalizations. Check carefully where exactly you are in phylospace, and do not rely on Genbank-only annotation.

*Answer: We combined the aprAB, dsrAB, mcrA and hydB-G phylogenetic trees into a new Figure (Fig. 7) that we implemented in the main text. We also provide an additional taxonomic tree based on rps3 transcripts as Supplementary Figure S4.*

*We also like to clarify that we did not sequence 16S rRNA gene amplicons. Further, the assignment of rps3 gene sequences at a deep taxonomic level requires the compilation of reference sequences into one's own database and to perform a HMM analysis. With regards to GTDB taxonomy, we consider that it is beyond the scope of the present manuscript to produce a detailed taxonomy from the MAGs obtained by our collaborators.*

[Figure]

--- end of review ---1

---

## Author Comment (AC2)

**Reviewer 2**

Schnabel and co-authors explored the impact of minor HC seepage on sediment sulfur cycling through generating a variety of datasets, including porewater geochemistry, reaction rate calculation, metagenomic and metatranscriptomic data. I admit that considerable efforts have been made to collect so many sediment cores and produce such comprehensive datasets. However, the present paper fails to formulate an informative and sound story; it reads more like a report rather than a scientific article. With that said, the data have not been deeply digested and synthesized. As a non-expert on microbiology, I can see that the microbiological data seem not to be well utilized and explored. The other flaw lies in the very limited novel perspectives provided by this study.

*Answer: We would like to thank the Reviewer for spending time and effort into revising our paper. We agreed to most of their suggestions to improve the paper.*

*Here, we provide point-by-point responses to the Reviewers' comments. Original comments by the Reviewer are listed in regular font while our answers appear in italic font.*

**Specific comments**

1. The title reads a little confusing. I don't see any discussion on the implication for HC reservoir detection.

*Answer: This topic was the initial driver for the study. We removed the second half of the title as we agree that the implications for HC detection are not discussed in detail.*

2. The Abstract is a little verbose; it is usually limited to one paragraph.

*Answer: The Abstract was condensed but we kept using paragraphs in the abstract as they make it easier to understand the content. Using paragraphs is not forbidden according to the journal's instructions to authors.*

3. HC positive site? How about HC bearing site?

*Answer: We changed all to HC sites.*

4. Line 94: The depth of SMTZ at ~100 m in the continental margin sediments is uncommon.

*Answer: This is right. It is the maximum depth and not the general depth. We revised the sentence.*

5. Line 154: what molecular analyses?

*Answer: Molecular analyses were specified as DNA/RNA in the text.*

6. Figures 2&3: I would suggest plotting the measured data instead of the extrapolated profiles. Maybe pick several representative profiles rather than the whole dataset.

*Answer: We have combined Figures 2, 3 and 4 and added some representative profiles to a new figure to show that the linear regression is justified. The new figure was implemented to the main text (Fig. 2). We believe that the trends we report in the main text can be captured much more quickly by depicting regression lines plotted jointly. We have also plotted the data points together with their regression lines, but the resulting plot appears too busy and one can no longer tell the individual cores apart (see plot here under).*

[Figure]

7.      Figure 4 can be combined with Figures 2&3.

*Answer: Done accordingly.*

8.      Figure 6. As CH4 and CO2 are not key data and they are subject to sampling artefact, they can be moved to the supplement.

*Answer: We beg to disagree with the reviewer as we feel that these data are necessary to put the other data into a context. For example, the very low methane concentrations clearly indicate that AOM is not a relevant process over the cored depth intervals.*

9.      Line 448: Higher occurrence? Weird wording. Also I don't see higher sulfate reduction rate in the deeper sediments from Fig. 5.

*Answer: There is a misunderstanding here. SR occurs more frequently at greater depths, but the sulfate reduction does not show higher sulfate reduction rates. We have reworded the sentence to avoid misunderstanding.*

Line 508: No sulfate profile in Figure 5.

*Answer: The wrong figure was mentioned. Corrected*

10.     Line 514: It is more likely that the lower alkalinity flux is attributed to HCO3- removal by authigenic carbonate precipitation, which can be demonstrated by Ca and Mg data if they are available.

*Answer: We assume that carbonate ions are originating from AOM at greater depths, diffusing upward and eventually precipitate as calcium carbonate. However, given the sulfate concentration profiles and the scarce methane data, it becomes clear that AOM is not a quantitatively important process in the cored depth interval. Also, the low level of detection of transcripts related to AOM suggest that the imprint on the geochemistry is minimal to non-detectable.*

Line 523: I think the latter hypothesis is more likely. I doubt the reoxidation of sulfide can contribute large variation of sulfate flux. My understanding is that sulfate derived from the reoxidation of sulfide is rapidly used via sulfate reduction rather than remaining in the porewater.

*Answer: We agree that the latter hypothesis is more likely, but believe that both hypotheses have their justification, nonetheless. We discuss both hypotheses now.*

---

## Author Response (AR2)

**Reviewer 1**

The revised manuscript is generally acceptable, but some grammatical mistakes and errors (mixing up carbonate and bicarbonate in lines 530 ff) need to be corrected. See details below.

*We would like to thank Prof. Teske for taking the time to review our paper again. We agree with his suggestions to improve the manuscript. Below are our point-by-point responses to his comments. The original comments are in regular font, and our responses are in italics.*

Figure 2:

Are you sure that panel 2f (showing actual profiles) will appear in clear resolution?

***Answer****: We agree that the resolution of Panel 2f, which displays the actual profiles, was insufficient in the submitted revision. To address this, we are now providing the figure in a higher quality, ensuring clearer and more detailed visuals.*

Line 530 ff:

One sentence defines alkalinity as bicarbonate (HCO3-) concentration; but the next sentence talks about carbonate (CO32-) as the main contributor to seawater alkalinity but from its context really means bicarbonate. Please correct this.

"We established ratios of sulfate to alkalinity (HCO3-) fluxes (Fig. 5). As about 90% of seawater alkalinity can be contributed to carbonate (89.8% HCO3-, 2.9 % CO32-) (Kerr et al., 2021) we set alkalinity synonymous with carbonate concentration."

***Answer:*** *We appreciate your attention to this and have corrected the mentioned section, and we also checked the remaining manuscript accordingly. The revised sentence reads:*

"We established ratios of sulfate to alkalinity ($HCO_3^-$) fluxes (Fig. 5). As about 90% of seawater alkalinity can be contributed to bicarbonate (89.8% $HCO_3^-$, 2.9 % $CO_3^{2-}$) (Kerr et al., 2021)we set alkalinity synonymous with bicarbonate concentration."

Line 755: "suggests is" does not fit here in this form; was a longer sentence intended?

***Answer:*** *You are right. We missed deleting a part of the former formulation, and we have now corrected it in the manuscript.*

Line 765: "scarce methane data" means that very limited information about methane is available (few data points and measurements). You probably want to say that only small amounts of methane were detected. It is possible to have plenty of methane data, but they indicate that only small amounts of methane exist.

***Answer:*** *We agree with your comment regarding the term 'scarce methane data. To better reflect the findings, we have revised the wording to indicate that only small amounts of methane were detected in the samples, rather than suggesting that limited data were available.*

**Reviewer 3**

General comments

The manuscript by Schnabel et al. presents an impressive set of pore-water data and sulfate reduction rates as well as metagenomic and metascriptomic data for 50 (40?) sediment cores/gravity cores retrieved from the SW Barents Sea shelf. The objectives of the study are to assess how the magnitude of upward methane/hydrocarbon flux impacts the geochemistry, biogeochemical processes – namely microbial sulfate reduction - and microbial communities in the surface and subsurface sediments.

The manuscript is definitely of interest for the readers of Biogeosciences and generally well-written. However, there are several issues that need to be specified and described much more precisely – in particular the terms „seepage" and „HC reservoir". There are also several statements and concepts presented in the manuscript that are not correct as given (e.g. statements about seismics). The authors should definitely define the term „seepage" and say what they mean when they speak of „seepage". What about the activity/episodicity of any potential seepage? Please, precisely specify and distinguish whether you speak of transport of methane by molecular diffusion (as is obviously the case at most of your sites) or fluid seepage – i.e. migration of fluids and or free gas bubbles through the pore space of the sediments at rates exceeding those of molecular diffusion. In other words, if upward methane transport occurs in the form of molecular diffusion – as seems to be the case at most of your study sites – I would not speak of seepage. I would therefore suggest to more generally speak of upward methane „fluxes" throughout the manuscript. The different intensities/magnitudes of methane upward flux then determine the depth position of the SMT and the magnitude of SR as well as the type of microbial community/ies.

I also did not fully understand which type of „HC reservoir" precisely you speak of. This is also not clear from Chapter 2. Do you mean free gas in the deeper subsurface? What about gas hydrates? The potential role of the presence of gas hydrates in the subsurface – as an intermediate methane reservouir - has not been mentioned and discussed at all. There are numerous studies that have demonstrated that during active seepage events methane is transported upwards from deeper sources (mostly in the form of free gas) and becomes trapped in the form of gas hydrates at shallower sediment depth (if positioned within the gas hydrate stability zone). After these gas hydrate deposits have formed they give off methane, which diffuses upward towards the sediment surface and leads to the establishment of an SMT where AOM consumes most of the upward diffusing methane (e.g., Dickens, 2001, GCA; Lapham et al., 2010, EPSL). The methane gradient – thus magnitude of upward flux – and depth position oft he SMT then depends on the depth position of the gas hydrates.

Moreover, the referencing to previous relevant studies is also not sufficient. In the past 20 to 25 years numerous studies have been performed to investigate the regional variability of upward methane fluxes. There are for example several studies by the group of Gerald Dickens that have investigated differences in upward methane fluxes – for example on Blake Ridge and in other ocean areas. Also the impact that upward methane fluxes and in particular of AOM on the geochemical composition of pore waters and sediments – including mineral dissolution (e.g. magnetite) and precipitation of authigenic minerals – including carbonates, barite, Fe sulfides/rock magnetic properties is mostly missing (see suggestions given below). There are also several previous studies that have correlated pore-water profiles with micobial communities (e.g. Oni et al., 2005, Frontiers

Microbiol.; Wunder et al., 2021, ISME; Schnakenberg et al., 2021, Frontiers Microbiol.).

It would also be good to have a zoom-in map of the study area in order to have an idea of the bathymetry and seafloor topography. The insert shown given in Fig. 1 b is not very informative. It would be good to see seafloor topography/bathymetry in ordert o assess whether there are typical seep seafloor features of methane seepage such as pockmarks and to find out at which water depth the study sites are located (also with respect to judging whether the sites lie within the gas hydrate stability or not).

*Answer: We thank the Reviewer for taking the time to revise our paper. We have agreed with most of the suggestions to improve the manuscript. Below, we provide point-by-point responses to the Reviewer's comments. The original comments are in regular font, and our responses are in italics.*

Specific comments

L. 24 and throughout the manuscript: The term „inconspicuous" is rather unusual in this context. Do you mean „low" upward fluxes?
*Answer: We rephrased the sentence for greater clarity and took out the term inconspicuous, by using "small and often unnoticed upwards HC fluxes"*

L. 30 Do you mean constant/no depletion instead of „linear"

*Answer: To clarify, by "linear profiles," we are referring to regression lines through measurement points that demonstrate a clear trend of either increasing or decreasing with depth. This explanation should address any confusion regarding the terminology. It's important to note that "linear" does not imply "no concentration change with depth." A quick image search for "linear profile" shows many examples with values that increase or decrease, confirming that the term is appropriately used. As such, the text remains unchanged, as we believe the term is sufficiently clear.*

L. 40: What precisely do you mean with „inconspicuous HC seepage" ? I would rather speak of low methane „fluxes". i.e. diffusive flux.

*Answer: similar to L24, was changed and is now defined*

L. 41: „shallower" than what precisely?!

*Answer: When we refer to "shallower" we specifically mean that sulfate depletion occurs at depths that are shallower than those typically observed in comparable, non-affected sites (i.e., sites not impacted by hydrocarbon (HC) seepage). This shift in depth is a direct consequence of the HC seepage, which influences the biogeochemical gradients in the shallow subsurface. To improve clarity, we have revised the wording at this point.*

L. 46: What precisely is a „minor" seep ?

*Answer: In response, we have added clarifying terms to the sentence to better explain the nature of minor seeps. Specifically, we now describe these seeps as "characterized by low, primarily diffusive HC fluxes" to make the distinction clearer. We hope this improves the clarity of the statement.*

L. 48: The effect of upward diffusion of methane and resulting AOM on sediment geochemistry has been shown and reported by numerous studies: including Riedinger et al. – and Henkel et al. etc.

**Answer:** *some references were added*

Ls. 49 to 51: These sentences are unclear. What precisely is a „distal manifestation"? „seabed"? I guess you mean sediment surface, right?!

**Answer:** *In response to your comment, we have rephrased the entire paragraph to avoid any misunderstandings.*

L. 52: I guess you mean „geochemical"  instead of geological, right?!

**Answer:** *Y*es, we agree—it was a mistake, and we have changed 'geological' to 'geochemical' in the manuscript as suggested

L. 54 ff.: The statement – as it stands here – is not correct. It is not methane-containing fluids that produce seismic signals but the presence of free gas that induces an impedance contrast that is registered by seismics. Fluids of high dissolved methane concentrations are not detectable by seismic approaches. Please, rephrase this more precisely throughout the manuscript.

**Answer:** *Thank you for your valuable feedback. We agree with your comment that it is the presence of free gas, rather than methane-containing fluids, that produces the seismic signals. We have rephrased the manuscript accordingly to more precisely reflect this distinction.*

L. 58 ff.: Strictly speaking, it is not the flux of methane itself but the consumption of methane/HCs in the process of AOM that impacts the geochemical composition of pore-waters and sediments. This has already been demonstrated by numerous studies, some of which should definitely also be refered to/cited here. So, in addition to what you describe/refer to in this part of the introduction you should at leasdt also mention a few of the most prominent impacts of upward methane flux and the resulting oxidation of methane by sulfate (AOM) . namely the precipitation of authigenic carbonates and barite (e.g. studies by Bohrmann, Torres, etc.) and also the dissolution of magnetite producing distinct minima in magnetic susceptibility (e.g. Riedinger et al., 2005; März et al., 2008).

**Answer:** *We agree that metabolic reactions and their products are primarily responsible for the majority of changes, and we have updated the text accordingly. However, we did not delve into the prominent manifestations of HC seepage at this stage of the introduction because we believe that if seepage is low, the larger manifestations will also be minor. We discuss AOM and carbonate precipitation in later sections of the introduction and the discussion. Therefore, we made only minor adjustments to the wording at the beginning of the paragraph and have also decided to include some relevant references at this point.*

L. 62/63: I also do not agree with this statement …. Methane formation is extremely widespread in continental margin sediments – and in most cases is not associated with underlying HC reservoirs but rather formed in situ by biogenic processes.
**Answer:** *Unfortunately, we cannot follow your problem. We cannot see the connection of your argument (which is correct) to our text.*

*We agree that the relative importance of sulfate reduction can vary depending on the specific environmental context. In response, we have clarified in the manuscript under which conditions sulfate reduction is the most important anaerobic organic matter degradation process. Additionally, we have incorporated several recent studies.*

L. 70: This is not entirely true … rather depends where you are. There are also more recent studies on the role of sulfate reduction (e.g. Bowles et al., 2014).

*Answer: We agree that the relative importance of sulfate reduction can vary depending on the specific environmental context. In response, we have clarified in the manuscript under which conditions sulfate reduction is the most important anaerobic organic matter degradation process. Additionally, we have incorporated several recent studies.*

L. 90: Please, also give reference to other relevant previous studies – e.g. Niewöhner et al. (1998; GCA), Treude et al.; Riedinger et al. (2005, 2014, 2017), März et al. (2008), Henkel et al. (2012; GCA).

*Answer: Thank you for your helpful suggestion. As recommended, we have added additional references to strengthen the manuscript.*

L. 113: This statement contradicts that in the abstract. Here you speak of 40 gravity cores while in the Abstract you mention 50 gravity cores.

*Answer: We have adjusted the sentence. There are 50 cores in total, 40 from HC-affected sites and 10 from reference sites*

Figure 1: The zoom-in in Fig 1b is not informative at all. It would be good to have a map showing the bathymetry/seafloor topography. The map also does not indicate where the potential „HC reservoirs" are found in the deeper subsurface.

*Answer: We added a bathymetric map to the figure.*

Figure 2 is very difficult to read and understand. What precisely is shown in Figs, 2a to 2d? These are definitely not measured pore-water profiles … are these modelled profiles or gradients? Please, specify and overhaul this figure as well as the figure caption.
The title of this figure says „sulfite" ?! I guess you mean sulfide, correct?!

*Answer: You are correct that the figures in Figs. 2a to 2d do not represent measured pore-water profiles. These plots show the modeled profiles (or modeled gradients) based on the linear regression analysis applied to the measured data. For a better understanding we changed the figure caption. We also corrected the spelling mistake for sulfide.*

---

## Author Response (AR3)

**Response to Editor's comments**

Dear Prof. Treude,

Thank you very much for handling our manuscript, the comments of the three reviewers were very constructive. Below is a response to your comments. Regarding the figures, we will now upload vectorized pdfs or high resolution (>300 dpi) images, so the production office can check whether the resolution is sufficient to see the small details, esp. in fig. 2.

On behalf of all authors, kind regards

Ellen Schnabel

**L47:** Another indicator of HC seepage can be chemosynthetic communities, such as sulfur bacteria mats, clams, and tubeworms.
*Sentence was updated accordingly.*

**Line 121:** Suggestion similar to above (chemosynthetic communities).
*Sentence was updated accordingly.*

**L148:** Weblinks always carry the risk of expiring or being moved. Can you provide more stable information for this mapping program? At a minimum, please provide the last access date of the website.
*We have reduced the use of links as much as possible and now cite literature instead.*

**L208:** Porosity is actually unitless (volume divided by volume).
*True. Thanks for catching this.*

**L272:** Same comment as above regarding weblinks. Please provide permanent information or (at a minimum) the last access date of the website.
*See above.*

**L287:** Same comment as above regarding weblinks. Please provide permanent information or (at a minimum) the last access date of the website.
*See above.*

**L307:** Same comment as above regarding weblinks. Please provide permanent information or (at a minimum) the last access date of the website.
*See above.*

**Fig 2f:** The resolution of this graph may be too low. I fear it will not be readable in the final paper. I consider the information important, but perhaps we should move it to the Supplementary materials in a larger size. Please collaborate with the production office to find the best solution.
*We have attempted to improve the image quality. All attempts to improve the image quality in Word were unsuccessful. However, we can provide the figure as a high-resolution PDF.*

**L327-328:** '…and some representative profiles.' I suggest changing it to: '…along with some representative profiles.'
*Done.*

**L555-556:** Can you specify the source of the information about anoxic conditions (or just cite the previous study)?
*The sentence was changed to "form other fieldwork in the area,…".*

**Data Availability:** Not a requirement since you provide the data in the Supplementary materials, but I think you should consider submitting the profiles to PANGAEA. The scientific community will likely appreciate it. :-) Just a recommendation.
*The data have been submitted to PANGAEA, but we have not received a DOI number yet. In case it arrives before publication we will inform the production office.*